

# Constraining the aerosol influence on cloud liquid water path

Edward Gryspeerdt[1], Tom Goren[2], Odran Sourdeval[2], Johannes Quaas[2], Johannes Mülmenstädt[2], Sudhakar Dipu[2], Claudia Unglaub[2], Andrew Gettelman[3], and Matthew Christensen[4]

[1]Space and Atmospheric Physics Group, Imperial College London, UK
[2]Institute for Meteorology, Universität Leipzig, Germany
[3]National Center for Atmospheric Research, Boulder, USA
[4]Department of Physics, University of Oxford, UK

**Correspondence:** E. Gryspeerdt
(e.gryspeerdt@imperial.ac.uk)

**Abstract.** The impact of aerosols on cloud properties is one of the largest uncertainties in the anthropogenic radiative forcing of the climate. In recent years, significant progress has been made in constraining this forcing using observations, but uncertainty still remains, particularly in the adjustments of cloud properties to aerosol perturbations. Cloud liquid water path (LWP) is the leading control on liquid-cloud albedo, making it important to observationally constrain the aerosol impact LWP.

Previous modelling and observational studies have shown that multiple processes play a role in determining the LWP response to aerosol perturbations, but that the aerosol effect can be difficult to isolate. Following previous studies using mediating variables, this work investigates use of the relationship between cloud droplet number concentration ($N_d$) and LWP for constraining the role of aerosols. Using joint probability histograms to account for the non-linear relationship, this work finds a relationship that is broadly consistent with previous studies. There is significant geographical variation in the relationship, partly due to role of meteorological factors (particularly relative humidity) in the relationship. However, the $N_d$-LWP relationship is negative in the majority of regions, suggesting that aerosol induced LWP reductions could offset a significant fraction of the radiative forcing from aerosol-cloud interactions (RFaci).

However, variations in the $N_d$-LWP relationship in response to volcanic and shipping aerosol perturbations indicate that the $N_d$-LWP relationship overestimates the $N_d$ impact on LWP. As such, the estimate of LWP changes due to aerosol in this work provides an upper bound to the radiative forcing from aerosol-induced changes in the LWP.

## 1 Introduction

Atmospheric aerosols are known to affect the radiative balance of the atmosphere, both through a direct interaction with radiation and via indirect interactions with cloud properties (Boucher et al., 2013). As almost all liquid cloud droplets form on an aerosol particle, changing the number and composition of aerosol particles can change the concentration of cloud droplets ($N_d$) in a cloud, leading to changes in the cloud brightness (Twomey, 1974) and possibly also leading to changes in the cloud fraction (CF or $f_c$) and liquid water path (LWP or $L$) through a delay in precipitation formation (eg. Albrecht, 1989). Estimates of radiative forcing due to changes in cloud properties vary significantly between different global climate models (Zelinka et al., 2014; Heyn et al., 2017), highlighting the need for observational constraints on the impact of aerosol on cloud properties.





Unlike greenhouse gases, aerosol properties vary strongly in space and time. This co-variation of aerosol and cloud properties in the present day atmosphere has been used to infer the impact of aerosols on cloud properties (e.g. Sekiguchi et al., 2003; Kaufman et al., 2005; Koren et al., 2005). Such observed relationships have been used to estimate the instantaneous radiative forcing (RFaci) from a change in N$_d$ (e.g. Quaas et al., 2008; Stevens et al., 2017; McCoy et al., 2017; Gryspeerdt et al., 2017) and of the aerosol induced change in CF (Chen et al., 2014; Goren and Rosenfeld, 2014; Gryspeerdt et al., 2016; Christensen et al., 2017). As the leading order term for determining cloud albedo (Engström et al., 2015), it is also vital to constrain aerosol effects on the in-cloud liquid water path (LWP), separate from changes in the CF. Existing studies show a mixed picture; while some model (Quaas et al., 2009; Seifert et al., 2015; Grosvenor et al., 2017; Neubauer et al., 2017) and observational studies (Gryspeerdt et al., 2014b; McCoy et al., 2018) suggest an increase in LWP with increasing aerosol, other studies (Wang et al., 2003; Small et al., 2009; Chen et al., 2014; Michibata et al., 2016; Christensen et al., 2017; Sato et al., 2018) find a reduction in LWP as aerosol increases. Some studies find both an increase and a decrease in LWP, depending on the meteorological conditions (Han et al., 2002; Ackerman et al., 2004; Bretherton et al., 2007; Xue et al., 2008; Toll et al., 2017; Bender et al., 2018), while other studies suggest a very weak LWP response to aerosol (Wang et al., 2012; Malavelle et al., 2017). The main aim of this work is to reconcile these previous studies and develop a constraint on the aerosol impact on LWP.

## 2 Isolating an aerosol effect

The key difficulty in interpreting observed aerosol-cloud relationships is separating the causal impact of aerosols (the change in LWP caused by an aerosol perturbation) from the confounding role of local meteorology (e.g. Quaas et al., 2010) and retrieval errors (e.g. Várnai and Marshak, 2009). Relative humidity in particular has been shown to obscure the causal relationship between aerosol optical depth (AOD) and CF (Quaas et al., 2010; Chand et al., 2012; Grandey et al., 2013). As many cloud properties are correlated to CF, the factors that obscure the aerosol-CF relationship can also confound other aerosol-cloud relationships, even those involving "intrinsic" cloud properties (Chen et al., 2014), such as cloud top pressure (Gryspeerdt et al., 2014a), and LWP (Christensen et al., 2017; Neubauer et al., 2017). Recent work (Gryspeerdt et al., 2016) has suggested that the use of a mediating variable such as N$_d$ can be used to account for the confounding influence of relative humidity. Following from this, the potential of the N$_d$-LWP relationship to constrain the aerosol impact on LWP is investigated in this work.

Similar to the aerosol-LWP relationship, where both potential aerosol effects and confounders can influence the strength of the relationship, several effects may influence the observed N$_d$-LWP relationship.

E1 **Aerosol effects** An increased aerosol concentration is likely to increase N$_d$. This increase in N$_d$ may affect cloud processes and in turn modify the LWP. There are several hypothesised pathways for a causal effect of aerosol on LWP:

(a) Precipitation suppression (Albrecht, 1989) - an increased N$_d$ at initially unchanged LWP implies reduced cloud droplet sizes, suppressing the formation of precipitation. This reduction in the cloud water loss to precipitation could subsequently increase cloud depth (Pincus and Baker, 1994) and thus LWP. While it has been demonstrated



that a reduction in droplet size suppresses precipitation (Suzuki et al., 2013), it is not clear how strongly this impacts LWP.

(b) The sedimentation-entrainment feedback (Ackerman et al., 2004; Bretherton et al., 2007) - the reduction in droplet radius from increased $N_d$ reduces the sedimentation flux in stratiform clouds, concentrating liquid water in the entrainment zone at the cloud top and increasing cloud-top evaporative and radiative cooling, increasing the entrainment rate. This increases the evaporative cooling in a positive feedback that depends on the above-cloud relative humidity, with drier air above cloud tops implying a larger LWP decrease. Negative $N_d$-LWP relationships in recent observational studies were suggested to have been due to this effect (Chen et al., 2014; Michibata et al., 2016; Sato et al., 2018).

(c) Evaporation-entrainment feedbacks (Wang et al., 2003; Xue and Feingold, 2006; Jiang et al., 2006; Small et al., 2009) - smaller droplets have a faster evaporation timescale, enhancing the cooling and hence the negative buoyancy at the edge of cumulus clouds. This intensifies the horizontal buoyancy gradient, increasing entrainment and hence evaporation, reducing the LWP with an expected similar meteorological dependency to E1b. Aircraft observations have found increased horizontal buoyancy gradients and reductions in cloud liquid water content (LWC) in polluted clouds (Small et al., 2009).

E2 **Retrieval errors** The MODIS LWP and $N_d$ both depend on the retrieved cloud top droplet effective radius ($r_e$) and cloud optical depth ($\tau_c$) and involve assumptions of varying validity (e.g., Grosvenor et al., 2018). Random errors in the

(a) Random errors in the retrieval of cloud properties ($\tau_c$, $r_e$) becoming correlated errors in LWP and $N_d$. Using $N_d$ and LWP calculated using the adiabatic assumption, random errors in $\tau_c$ will generate a positive $N_d$-LWP sensitivity ($\frac{d \ln L}{d \ln N_d} = 2$), while errors in $r_e$ will generate a negative sensitivity ($\frac{d \ln L}{d \ln N_d} = -0.4$), see appendix A for details.

(b) Sub-adiabatic clouds. Both the LWP and the $N_d$ retrieval make assumptions about the adiabaticity of clouds. Variations in the adiabaticity (Merk et al., 2015), even across a single cloud can therefore generate a positive $N_d$-LWP sensitivity ($\frac{d \ln L}{d \ln N_d} = 2$).

E3 **Feedbacks** A change to the LWP may affect $N_d$, obscuring the causal impact of $N_d$ on LWP. This feedback may depend on other meteorological parameters, generating an apparent dependence on local meteorology in the observed $N_d$-LWP relationship. The existence of strong feedbacks can make using a mediating variable to account for meteorological covariation problematic (Pearl, 1994).

(a) Precipitation preferentially occurs at large LWP. Precipitation scavenging of aerosol can reduce the amount of aerosol available for future activation to cloud droplets, reducing $N_d$. Conversely, if an increased $N_d$ decreases the precipitation rate, this could result in a further increase in the $N_d$ through a reduction in wet scavenging and an increase in the available aerosol (a positive feedback).




(b) The impact of entrainment on the retrieved $N_d$. The impact of entrainment on $r_e$ depends on the mixing type. Extreme inhomogeneous mixing (Baker et al., 1980) leads to a reduction in $N_d$ and LWP, but no immediate change in the droplet size distribution or retrieved $N_d$. In contrast, homogeneous mixing (Warner, 1973) results in a reduction in $r_e$ and so an increase in the retrieved $N_d$. A larger change in retrieved $N_d$ (and LWP) would be expected for increased dry air entrainment, generating a negative $N_d$-LWP relationship as the LWP changes.

E4 **Additional confounders** Although using $N_d$ as a mediating variable helps to account for the impact of RH on the aerosol-LWP relationship, additional meterological confounders, impacting both $N_d$ and LWP may still impact the $N_d$-LWP relationship, obscuring the causal impact of $N_d$ on LWP. An example case could be a convergence situation that leads to large moisture (large LWP) and large updraught (large $N_d$, even at constant aerosol).

These effects are depicted in Fig. 1. To constrain the causal aerosol influence on LWP, the impact of E1 has to be identified and isolated from that of E2-4. This would allow the aerosol impact on LWP to be constrained using the $N_d$-LWP relationship.

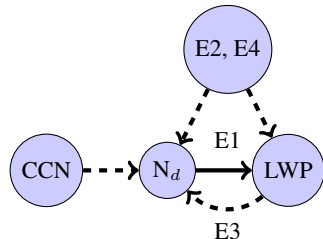

**Figure 1.** A simplified picture of the $N_d$-LWP system, showing factors impacting the causal relationship ("E1") - potential meteorological confounders and retrieval errors ("E2,E4"), LWP dependent controls on $N_d$ ("E3") and the impact of aerosols on $N_d$ (CCN).

It is necessary to understand the role of these different processes on the $N_d$-LWP relationship in order to determine the impact of aerosols on the LWP. Using a variety of different satellite retrievals along with reanalysis data, the $N_d$-LWP relationship is investigated globally and the impact of meteorology is explored. To understand the role of feedbacks (E3) and additional confounders (E4), natural experiments are used to examine the $N_d$-LWP relationship in regions where there is a strong aerosol perturbation. Finally, the observed relationship is converted to a radiative forcing, allowing it to be compared to other observational studies and to be used for further analysis of the aerosol impact on clouds and the climate.

## 3 Methods

This work is based on observational data from the Aqua satellite, specifically the moderate resolution imaging spectroradiometer (MODIS), the advanced microwave scanning radiometer-EOS (AMSR-E) and the clouds and the Earth's radiant energy system (CERES) instruments for a three year period (2007-2009 inclusive).

$N_d$ is retrieved using the level 2, collection 6, MODIS cloud property dataset (MYD06_L2) at a 1 km by 1 km resolution, making use of the adiabatic assumption (Brenguier et al., 2000; Quaas et al., 2006). Following the work of Grosvenor and



Wood (2014) and Bennartz and Rausch (2017), the $N_d$ is filtered to include only liquid, single layer clouds with a top warmer than 268 K at 1 km resolution. In addition, pixels with an optical depth smaller than 4 or an effective radius less than 4 $\mu$m are excluded due to the uncertainty of these retrievals (Sourdeval et al., 2016). Pixels with a 5 km cloud fraction less than 0.9 are excluded to remove pixels close to cloud edges, and only pixels with a solar zenith angle of less than 65° and a sensor

zenith angle of less than 41.4° are used to reduce the impact of known biases (Grosvenor and Wood, 2014; Eastman and Wood, 2016; Grosvenor et al., 2018). Finally, only pixels with an inhomogeneity index (Cloud_Mask_SPI) of less than 30 are used to account for biases in the effective radius ($r_e$) in inhomogeneous scenes (Zhang and Platnick, 2011). Trials using a more stringent upper limit of 10 show little difference to the results presented here (not shown). The $N_d$ is gridded to a 1° by 1° resolution and finally, the condensation rate temperature correction from Gryspeerdt et al. (2016) is applied.

The MODIS LWP is gridded to a 1° by 1° resolution from MYD06_L2, selecting only liquid, single layer clouds with tops warmer than 268 K. The extra filtering applied to the $N_d$ is not applied to the LWP at the pixel resolution as the LWP is less sensitive to $r_e$ biases and this filtering would significantly bias the LWP against AMSR-E by selecting primarily high LWP scenes. However, only 1° by 1° gridboxes with a $N_d$ retrieval are retained for this analysis, resulting in an implicit filtering by satellite and solar zenith angles.

As both the MODIS LWP and $N_d$ rely on the adiabatic assumption and the same retrieved cloud properties, there is a significant potential for errors in these properties due to failures of the adiabatic assumption (Merk et al., 2015) and consequent correlated errors generating a $N_d$-LWP relationship (E2b). The $N_d$ retrieval is better able to deal with non-adiabatic clouds than the effective radius retrieval alone (Painemal and Zuidema, 2011). For the majority of this work, the LWP is determined using V6 of the AMSR-E Ocean product (Wentz and Meissner, 2004), a passive microwave product that does not depend on

the adiabatic assumption. Clear-sky bias corrections are applied following Lebsock and Su (2014) at the pixel level. As the windspeed and sea surface temperature retrievals are unreliable in precipitating scenes, they are interpolated to precipitating locations by fitting a cubic mesh (Jones et al., 2001). To determine the in-cloud LWP, the AMSR-E LWP is divided by the MODIS cloud retrieval cloud fraction (CF) at the AMSR-E pixel level (14 km), with pixels having a CF of less than 10% being excluded due to the large uncertainty in the resulting in-cloud LWP. Finally, the AMSR-E data is gridded from the sensor

footprint of 14 km to a 1° by 1° resolution.

As a linear sensitivity ($\frac{d \ln L}{d \ln N_d}$) is not able to fully describe the non-linear relationship between $N_d$ and LWP, a piecewise relationship of the form (Eq.1) is used. $L^p$ and $N_d^p$ are the LWP and $N_d$ values at the intersection between the two parts of the curve, while $m_l$ and $m_h$ are the gradients of the fit for the low and high $N_d$ portions of the curve. This curve is fit to the $N_d$-LWP joint probability histogram ($P(L|N_d)$), using the Levenberg-Marquardt algorithm in log-space (Jones et al., 2001). By

fitting to the joint probability histogram, each $N_d$ bin is given equal weight, rather than the weighting by the present day $N_d$ probability distribution implicit in the standard linear regression.

$$\ln L = \ln L^p + m_l \left( \ln N_d - \ln N_d^p \right) \qquad\qquad N_d < N_d^p$$
$$\ln L = \ln L^p + m_h \left( \ln N_d - \ln N_d^p \right) \qquad\qquad N_d \geq N_d^p \qquad\qquad (1)$$



To convert a change in LWP to a change in top of atmosphere radiation, data from the CERES 1 degree daily Single Scanner Footprint, Edition 4 dataset is used (Wielicki et al., 1996). The all-sky albedo from CERES ($\alpha$) is histogrammed as a function of the CF ($f_c$), LWP and $N_d$, creating a single, global, joint probability histogram ($P(\alpha|f_c, L, N_d)$). Given the retrieved cloud properties for a location ($f_c$, LWP and $N_d$), this histogram produces a distribution of consistent values of the all-sky albedo

($P(\alpha)$). This can be used to calculate the mean oceanic albedo to within 1% in the tropics, with an RMS error in the tropics of 1%, increasing to around 5% near the poles. These variations are primarily due to differences in the mean solar zenith angle between the MODIS and CERES datasets, such that they have a small effect when determining the albedo sensitivities in this work.

Following Eq. 2, the $N_d$-LWP and $N_d$-$f_c$ relationships can be used to determine a change in scene/all-sky albedo as a function

of an $N_d$ change. The relationships are treated as conditional probabilities ($P(L|N_d) = \frac{P(L,N_d)}{P(N_d)}$), following Gryspeerdt et al. (2016). When combined with the $N_d$ sensitivity to aerosol ($\tau_a$) changes $P(N_d|\tau_a)$, this allows the scene albedo as a function of aerosol ($P(\alpha|\hat{\tau_a})$) to be calculated for a given scene of liquid clouds (Eq.3), where the circumflex indicates that a variable has been set to a certain value (the causal relationship). Note that this is different from the observed relationship $P(\alpha|\tau_a)$, due to the confounding effects of local meteorology (Pearl, 1994; Gryspeerdt et al., 2016). It also makes the assumption that the observed

conditional probabilities represent the causal relationship (i.e. $P(L|N_d)$=$P(L|\hat{N_d})$, representing only E1), an assumption that will be investigated in this work.

$$P(\alpha|\hat{N_d}) = \sum_{f_c} \sum_{L} P(\alpha|f_c, L, N_d) P(f_c|N_d) P(L|N_d) \tag{2}$$

$$P(\alpha|\hat{\tau_a}) = \sum_{N_d} P(\alpha|\hat{N_d}) P(N_d|\tau_a) \tag{3}$$

The albedo sensitivity to aerosol through modifications of each of the components of the albedo ($N_d$, L, $f_c$) can be determined by replacing probabilities conditioned on $N_d$ with unconditional probabilities. For example, the sensitivity due only to $N_d$ variations (the Twomey effect; Twomey, 1974) can be determined by removing any dependence of CF and LWP on $N_d$ ($P(f_c|N_d) = P(f_c)$ and $P(L|N_d) = P(L)$) in Eq. 2. The change in planetary albedo is then determined by multiplying each gridbox by 1-$f_c^{ice}$ (the ice cloud fraction), making the implicit assumption that there is no change in the ice cloud albedo or

$f_c^{ice}$. This is converted to a radiative forcing by multiplying by the anthropogenic aerosol fraction from Bellouin et al. (2013) and the incoming solar flux.

To avoid uncertainties associated with the aerosol anthropogenic fraction inherent in estimates of the aerosol radiative forcing, the ERF due to LWP changes is not reported directly, only as a fraction of the RFaci (Bellouin et al., in prep). The value for the forcing due to LWP changes can be re-constructed using an appropriate estimate of the RFaci if required (e.g. Quaas

et al., 2008; Stevens et al., 2017; McCoy et al., 2017; Gryspeerdt et al., 2017).



# 4   The $N_d$-LWP relationship

## 4.1   Global relationships

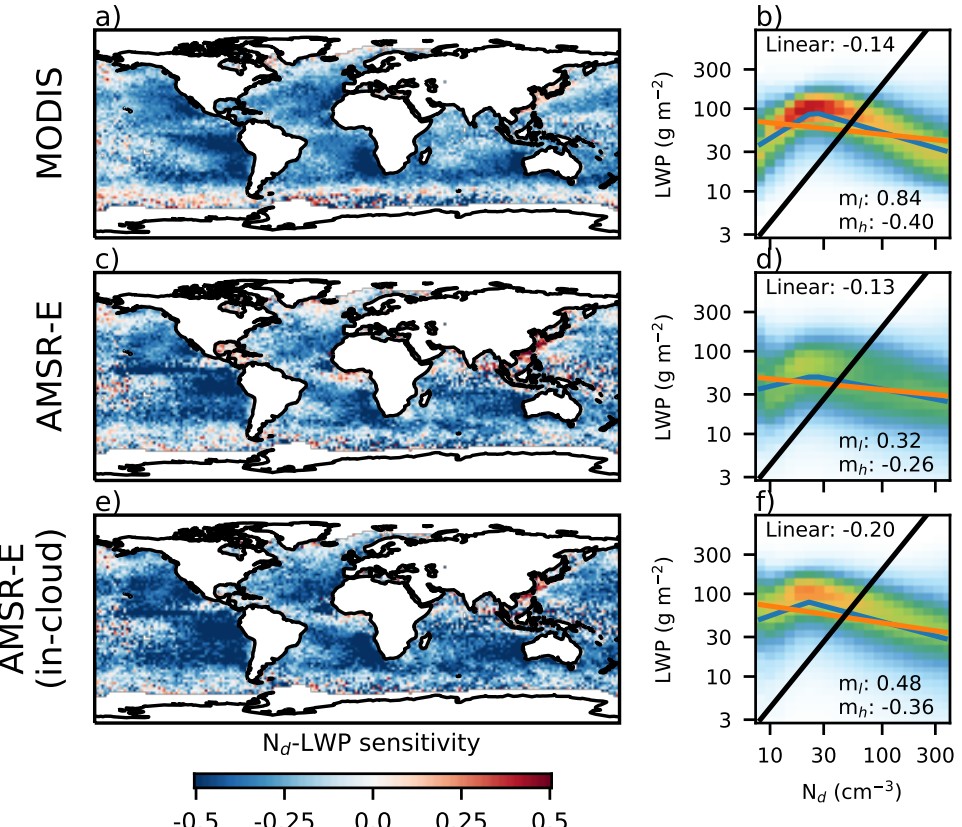

**Figure 2.** Left column: The sensitivity (linear regression coefficient in log-log space) of $N_d$ to LWP for a selection of different LWP measures, using MODIS $N_d$ for the period 2007-2009. The sensitivities are calculated at a $1°$ by $1°$ resolution from instantaneous (daily) data. a) MODIS LWP, b) AMSR-E (all-sky) LWP and c) AMSR-E (in-cloud) LWP. The right hand column shows the global $N_d$-LWP joint histogram, where each column is normalised so that it sums to one (showing P(LWP|$N_d$)). The black line is at an effective radius of $15\mu$m (assuming adiabatic clouds), an approximate indicator of precipitation, with precipitating clouds lying to the upper left of the line. The orange line as a linear regression on the data, with the linear sensitivity shown in the top left of the subplot. The blue line is a fit of the form Eq. 1, with the gradients $m_l$ and $m_h$ shown in the lower right of each subplot.

Similar to previous studies (Michibata et al., 2016), a negative linear $N_d$-LWP sensitivity (Fig. 2a, equivalent to the slope of the orange line in Fig. 2b) is found globally over oceans, with a particularly strong negative relationship in the subtropical stratocumulus decks off the western coasts of continents. Positive sensitivities are observed in some regions, particularly in the



East China Sea. The sensitivity becomes noisier close to the international dateline, due to a mismatch between the MODIS and AMSR-E definitions of a day.

A similar negative relationship is observed when using the AMSR-E LWP, both the all-sky LWP (Fig 2c) and the in-cloud LWP (Fig. 2e). The relationship in Fig. 2c, using the all-sky LWP, is much weaker than the in-cloud LWP in Fig. 2e, which is
the most strongly negative linear sensitivity of the three relationships in Fig. 2. A strong positive relationship remains in the East China Sea.

The $N_d$-LWP joint histograms shown in the right hand column of Fig. 2 show that the $N_d$-LWP relationship is highly non-linear at a global scale. All of the histograms show an increase in the LWP with increasing $N_d$ at low $N_d$, followed by a decrease in the LWP at high $N_d$. Despite the global variation in $N_d$, this non-linearity is not obvious in the global plots of the linear
sensitivity. However, a similar variation in the sensitivity simulated in LES (Xue et al., 2008) and in studies of shiptracks, where the impact of the injection of aerosol from shipping depends on the background cloud state (Goren and Rosenfeld, 2014; Toll et al., 2017). This non-linearity is consistent with the action of two proposed aerosol effects in liquid clouds (E1). The positive relationship at low $N_d$ is consistent with precipitation suppression, occurring only in the precipitating region of the $N_d$-LWP space (left of the black line in Figs. 2b,d,f). The negative relationship at high $N_d$, in regions of $N_d$-LWP space where the cloud
is unlikely precipitating (right of the black line), support the model-based results of Ackerman et al. (2004), where a high $N_d$ can result in a LWP reduction in clouds where precipitation does not reach the surface.

The differences between the fits of Eq.1 to the MODIS (Fig 2b) and the AMSR-E (Fig 2f) histograms demonstrate how a simple linear regression for calculating a sensitivity does not capture the strength or nature of the relationship. The AMSR-E relationship in Fig. 2f has a slightly weaker negative relationship at high $N_d$ ($m_h$) than that found using MODIS data (Fig. 2b),
but a 50% more strongly negative sensitivity worldwide. This shows the importance of considering the complete relationship and suggests that the linear sensitivity alone is not a strict constraint on the aerosol impact on LWP. The MODIS $N_d$-LWP relationship has an $m_h$ close to the value expected due to errors in the $r_e$ retrieval (-0.4). The $m_h$ values for the in-cloud LWP from both MODIS and AMSR-E are larger than those from the LES simulations of Ackerman et al. (2004) ($m_h \approx -0.2$ for the DYCOMS and dry ASTEX cases), Bretherton et al. (2007) (equivalent $m_h \approx -0.1$) and Xue et al. (2008) ($m_h < -0.2$).
The non-linear behaviour of the $N_d$-LWP relationship is similar to that expected due to correlated errors in the MODIS $N_d$ and LWP retrievals (E2, Appendix A). However, the similarity between the MODIS (Fig. 2b) and the in-cloud AMSR-E (Fig. 2f) relationships (unaffected by correlated errors due to the independent LWP measurement) shows that although correlated errors (E2) may play a role in determining the $N_d$-LWP relationship, they do not dominate it. However, to avoid any further impact of E2, the AMSR-E in-cloud LWP is used to characterise the $N_d$-LWP relationship for the remainder of this
work.

## 4.2 Regional relationships

Due to the difficulty of visualising joint histograms globally, a clustering approach is used to select regions with similar microphysics. A k-means clustering method (Anderberg, 1973) is used on the $N_d$-LWP joint probability histograms representing each 1° by 1°gridbox. The algorithm is modified to deal with missing data (k-POD; Chi et al., 2016), resulting in two distinct



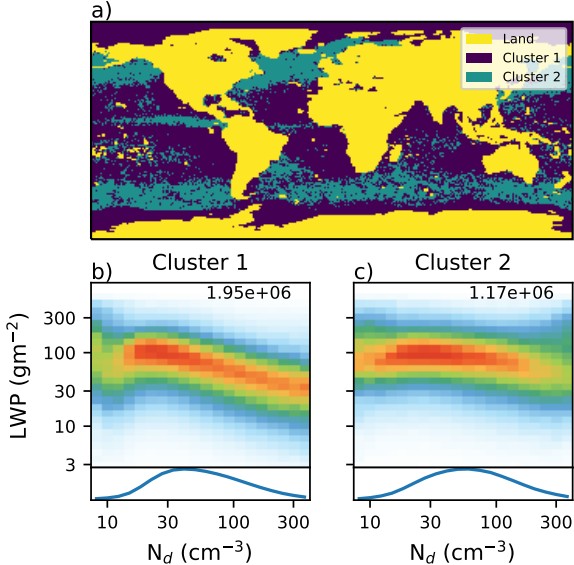

**Figure 3.** a) The location of the oceanic clusters for the $N_d$-LWP relationship, determined using the k-POD clustering method, using MODIS $N_d$ and the AMSR-E in-cloud LWP. b) and c) $N_d$-LWP joint histograms for the two clusters (as in Fig. 2) The line plot at the bottom shows the occurrence of each $N_d$ value for each cluster and the number of retrievals assigned to each cluster is displayed in the upper right of each histogram.

clusters over ocean with each gridbox being assigned to a single cluster (Fig.3). The first cluster (Fig.3b) is found primarily in the subtropical subsidence regions, particularly in the Pacific and South Atlantic. This cluster is characterised by an increase in LWP with $N_d$ at low $N_d$, followed by a decrease in LWP at high $N_d$, similar to the global relationships in Fig.2.

The second cluster (Fig. 3c) occurs mostly in the tropics and in the extratropics, regions with a much lower liquid CF. The $N_d$ distribution is less skewed towards lower values in this cluster. This cluster only includes about half the number of retrievals of the first cluster, occurring over a smaller area in regions that typically have a lower liquid CF. This lower frequency of occurrence explains the similarity of the global results with the first cluster.

The primary difference between the clusters is in their behaviour at high $N_d$. Whilst the subtropical cluster (1) shows a decrease in LWP with increasing $N_d$ (negative $m_h$), the second cluster is almost insensitive to $N_d$, even showing a slight increase in the LWP at the highest $N_d$ values. This may indicate a difference in the processes important for forming precipitation in the two different clusters (Mülmenstädt et al., 2015) and so a different response to $N_d$ perturbations. The weak sensitivity of LWP to $N_d$ (Fig. 3c) fits with the results of Malavelle et al. (2017), suggesting a weak response of LWP to $N_d$ variations focusing on a region where cluster two dominates. However, it means that the mid-latitude response may be a poor constraint on the response of the subtropical stratocumulus to $N_d$ perturbations, an issue that is of particular importance given the large role of the stratocumulus decks for the global aerosol forcing (Gryspeerdt and Stier, 2012).





## 4.3 The impact of meteorology

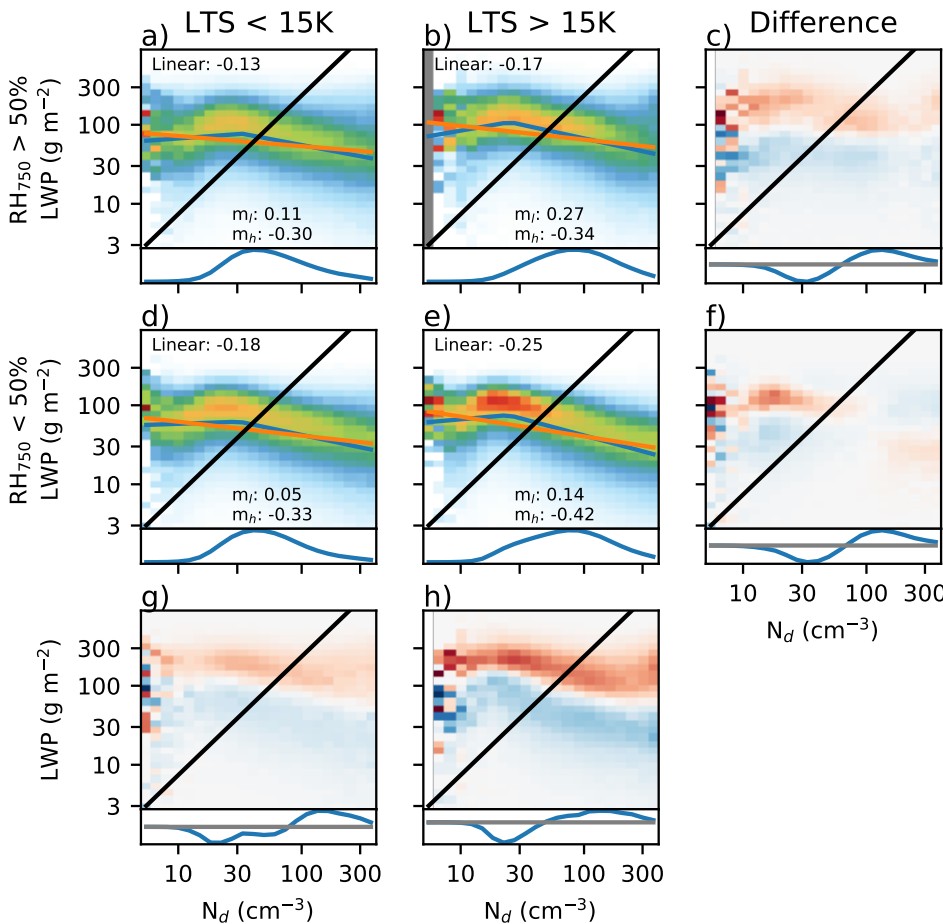

**Figure 4.** Joint histograms (as in Fig.2) created for meteorological conditions, separating by RH at 750 hPa and LTS. The difference plots are shown at the end of each row/column, with red above blue in each column showing an increase in AMSR-E in-cloud LWP at high LTS/RH$_{750}$ for a given N$_d$. The histograms under each joint histogram show P(N$_d$) for each set of meteorological conditions.

While the overall form of the relationship remains the same, there is some variation in the N$_d$-LWP joint histogram as a function of the meteorological state (Fig. 4). Following previous studies (Chen et al., 2014; Michibata et al., 2016), the data are separated by low troposphere stability (LTS) and relative humidity at 750 hPa (RH$_{750}$; approximately cloud top).

5    The response to LTS variations is small, occurring primarily in the part of N$_d$-LWP space where precipitation is expected (Figs.4c,f). The weak response to LTS is different from previous studies, which have shown a similar sized response to LTS and RH changes (Chen et al., 2014). A comparison between Figs. 4a,b shows that this variation in the linear sensitivity is partly




due to variations in the $N_d$ distribution. At high LTS (Fig. 4b), the mean $N_d$ is larger than that found at low LTS (Fig. 4a), resulting in a more negative linear sensitivity. However, the high-$N_d$ sensitivity from the fitted relationship ($m_h$) is very similar at both high and low LTS. The difference in the precipitating region sensitivity ($m_l$) may be due to variations in the precipitation processes or regime dependent retrieval errors for shallow cumulus (low LTS) and stratocumulus clouds (high LTS). However,
the low frequency of occurrence of these low-$N_d$ conditions (the histograms under each joint histogram in Fig. 4) limits their impact on the mean $N_d$-LWP sensitivity.

  The difference in $N_d$-LWP histograms for the two $RH_{750}$ classes is much more pronounced, particularly for the high LTS cases (Figs. 4b,e), where stratocumulus clouds are common. This may be due to the dependence of the evaporation-entrainment feedback (E1c) on cloud edge entrainment, where a weaker relationship to cloud top relative humidity might be expected than
in cases where the sedimentation-entrainment feedback (E1b; and hence cloud top entrainment) dominates. At high $N_d$, there is a significant shift in the LWP towards higher values with increasing $RH_{750}$, resulting in a decrease in the magnitude of $m_h$ as the $RH_{750}$ increases. A relative decrease in $m_h$ of around 20% is observed, slightly smaller than the 30% decrease in the linear sensitivity. Unlike the variations in the sensitivity with LTS, the increase in $N_d$ with increasing $RH_{750}$ is accompanied by a decrease in the linear sensitivity, showing that changes in the $N_d$ distribution are not the sole controller of the magnitude
of the linear sensitivity and that this measure of the relationship can provide information about $m_h$.

  These changes in $m_h$ as a function of $RH_{750}$ and LTS fit the conclusions of previous studies (Ackerman et al., 2004; Chen et al., 2014; Michibata et al., 2016); increased entrainment at higher $N_d$ results in a reduction of the LWP, with a stronger decrease at lower cloud top humidities. The resulting decrease in LWP with increasing $N_d$ would reduce cloud albedo, offsetting the RFaci (also due to an increase in $N_d$) and reducing the overall ERFaci.

## 5 Feedbacks and additional confounders

The strong negative relationship observed in Sec. 4 and in previous observational studies (Chen et al., 2014; Michibata et al., 2016; Sato et al., 2018) is in contrast to recent studies showing a weak or varied LWP response to aerosol perturbations (Chen et al., 2012; Christensen et al., 2014; Malavelle et al., 2017; Toll et al., 2017). While a negative $N_d$-LWP relationship has been found in some modelling studies with large-eddy simulations (Ackerman et al., 2004), the strength of this negative relationship
($m_h \approx$ -0.2) is weaker than the sensitivities observed in Sec. 4. It is possible that feedbacks (E3) or the existence of additional confounders (E4) could be obscuring the causal relationship (Fig. 1). This would reduce the utility of the $N_d$-LWP relationship as a constraint on aerosol-cloud interactions in climate models and for determining the aerosol radiative forcing.

  In situations where there is a loop or feedback in the causal graph (e.g. Fig. 1), an experiment is required to determine the strength of the causal relationship. Although the capability to artificially alter $N_d$ over a large spatial and temporal scale does
not exist, large aerosol perturbations are able to alter the CCN environment and hence $N_d$ independently of any feedbacks or confounders (E2-4; Fig. 1). The $N_d$-LWP relationship produced by these "natural experiments" would therefore be expected to be closer to the causal impact of aerosol on LWP than the relationship determined in Sec. 4.



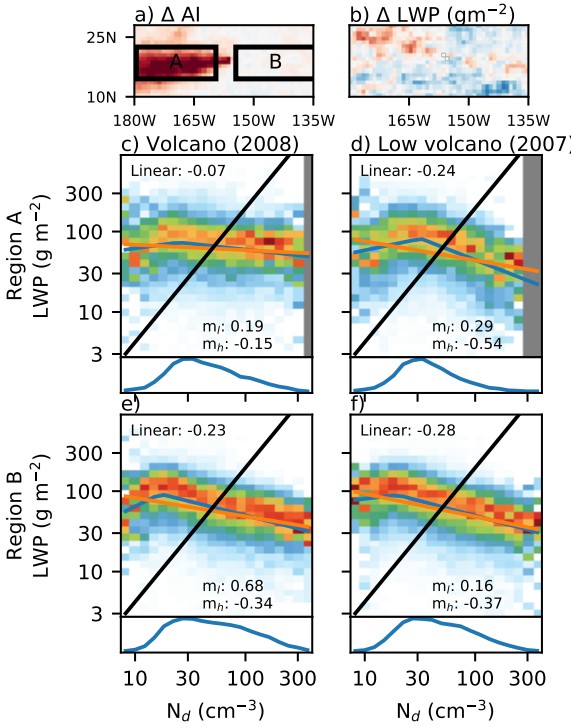

**Figure 5.** $N_d$-LWP relationships as in Fig. 2 in two regions around Hawai'i for two years, a low emissions year (2007) and a high emissions year (2008). (a) and (b) show the difference in AI and LWP between the high and low emission years, with red indicating an increase in 2008. (c-f) show the $N_d$-LWP joint histograms (as in Fig. 2) for the two periods in the regions from (a).

Volcanoes provide a possible natural experiment (e.g. Gassó, 2008; Yuan et al., 2011; Toll et al., 2017), as their $SO_2$ emissions are independent of the prevailing meteorological conditions (Gassó, 2008). Following Yuan et al. (2011), the Kilauea volcano on the island of Hawai'i is used as an exogenous aerosol perturbation. Previous work has shown a stronger linear AOD-$N_d$ sensitivity downwind of the Hawai'i than in surrounding regions, demonstrating the strong impact the $SO_2$ from Kilauea has on $N_d$ (Gryspeerdt and Stier, 2012). There is significant variability in the $SO_2$ emitted from the volcano. Comparing a year with strong $SO_2$ emissions (2008) with a low emissions year (2007) shows that the variation in aerosol index (AI; AOD times Ångström exponent; Nakajima et al., 2001) downwind from the volcano comes primarily from the variation in aerosol (Fig. 5a), rather than in meteorological conditions.

Despite the strong negative $N_d$-LWP relationship observed in sub-tropical regions (Fig. 3b), there is no change in the LWP (Fig. 5b) in the region with a strong change in AI (region A). This lack of a LWP response to volcanic emissions is similar to the results of Malavelle et al. (2017), but is within the area covered by the more sensitive cluster (Fig. 3). The weak LWP response





to aerosol variations suggests that the strong negative $N_d$-LWP relationship (Figs. 2, 3) is unlikely to describe the impact of $N_d$ variations on LWP.

This interpretation is supported by the variation of the $N_d$-LWP relationships as a function of $SO_2$ emissions. In 2007, volcanic emissions were weak and the $N_d$-LWP relationship was very similar between the regions downwind (region A; Fig. 5d) and upwind (B; Fig. 5f) of the volcano, with a strongly negative $m_h$ and negative linear sensitivity. However, in the high aerosol environment of 2008 (Fig. 5c), this negative relationship becomes much weaker in the volcanic plume ($m_h$=-0.15), whilst little change is observed upwind of the island (Fig. 5e). The lack of a change in region B indicates that the meteorological conditions were similar in both years, such that the changes in region A can be attributed to the aerosol variations (E1).

In the absence of feedbacks (E3), additional confounders (E4) and meteorological variations, the $N_d$-LWP relationship should be insensitive to the cause of the $N_d$ variations. Given the similarity in the meteorological conditions between the years, the difference in the $N_d$-LWP relationship in region A therefore suggests that the relationship is modified by feedbacks (E3) or additional confounders (E4). Due to the high volcanic emissions, the 2008 $N_d$-LWP relationship in region A is known to be strongly controlled by aerosol variations (E1) and has a reduced impact of other processes (E2-4), such that it is likely closer to the causal $N_d$-LWP relationship. This indicates a considerably weaker role for $N_d$ than determined in Sec. 4. With an $m_h$ of -0.15, the in-plume results are much closer to the results from LES simulations (Ackerman et al., 2004; Bretherton et al., 2007; Xue et al., 2008, $m_h < -0.2$) and in-situ observations of shiptracks, where decreases in LWP have been observed in particularly polluted conditions (Ackerman et al., 2000; Noone et al., 2000; Christensen and Stephens, 2011; Goren and Rosenfeld, 2014). The consequently weaker LWP response to aerosol is in better agreement with the weak LWP changes observed in Fig. 5b and Malavelle et al. (2017).

The Kilauea volcano affects primarily shallow cumulus clouds (Oreopoulos et al., 2014), which exert a weak control on the ERFaci from LWP changes due to their low liquid CF. The processes responsible for a reduction in LWP (E1c) may be different from those controlling stratocumulus clouds (E1b). Shipping provides another source of exogeneous aerosol perturbations, generating shiptracks that are primarily concentrated in the high CF stratocumulus regions. Using a database of shiptracks from Christensen et al. (2014), the relationship between the in-shiptrack $N_d$ and LWP increase in the shiptrack compared to the control region around the track (dLWP) indicates how the LWP responds to $N_d$ perturbations. As the $N_d$ values always increase from the control region to the inside the shiptrack, dLWP shares a sign with the gradient of the $N_d$-LWP relationship. Note that due to the required spatial resolution, the LWP for these shiptracks is retrieved using MODIS, rather than AMSR-E.

For low control values of the LWP (Fig. 6), increases in LWP (positive values of dLWP) are seen at lower in-shiptrack values of $N_d$, but as the shiptrack $N_d$ gets higher, the dLWP reduces to close to zero, with a negative dLWP for the most polluted cases. When the control LWP is high, dLWP is consistently weakly negative, although this likely is due to regression to the mean effects (the mean control LWP is $82\,\mathrm{g\,m^{-2}}$). This suggests that the LWP becomes insensitive to further aerosol/$N_d$ perturbations once the LWP reaches a sufficient magnitude, consistent with an aerosol suppression of precipitation (E1a). These small dLWP values at high $N_d$ are consistent with the Kilauea results, suggesting a weak LWP response at high $N_d$. If the LWP response in shiptracks followed the relationships from Sec. 4, a strong negative dLWP should be visible at high $N_d$, in contrast to the weak negative response actually observed (Fig. 6).



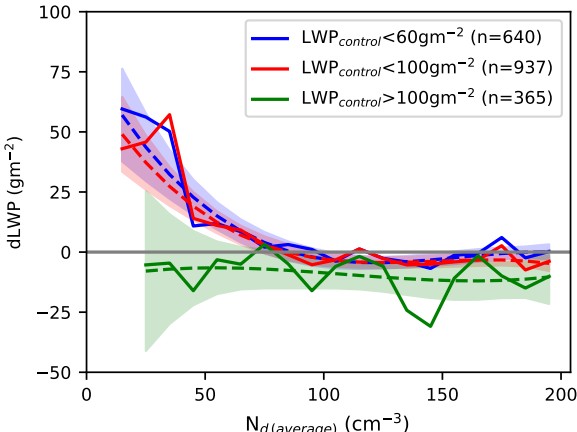

**Figure 6.** The difference in the LWP between the shiptrack and surrounding control regions as a function of the shiptrack $N_d$. The separate lines are for different values of control LWP. The LWP and $N_d$ values are from MODIS, using the shiptrack dataset from Christensen et al. (2014). The numbers in the legend are the number of shiptracks that makeup each line. Each line is characterised by a third-order uncertainty-weighted polynomial fit (dashed), with the shaded area showing the $2\sigma$ uncertainty on the fit.

By selecting situations where aerosol is known to be responsible for $N_d$ variations (so-called "natural experiments"), the impact of feedbacks (E3) and additional covariations (E4) can be reduced. In these situations, the $N_d$ variations are driven by exogeneous aerosol perturbations, such that the LWP variations are a response to (rather than a driver or indicator of) the change in $N_d$ (E1 only). This means that the $N_d$-LWP relationship during these "natural experiments" provides better information on

5    the LWP response to $N_d$ variations, such that the strong negative $N_d$-LWP relationships observed in Sec. 4 likely overestimate the decrease in LWP in response to aerosol perturbations. While the satellite-derived relationships may therefore be unsuitable as a direct estimate on the aerosol impact on LWP, they could be used as a lower bound on the LWP change (an upper bound on the radiative forcing) from aerosol-induced LWP decreases.

## 6    The implied ERFaci

10   The planetary albedo sensitivities to aerosol perturbations are shown in Fig.7 following Eq. 2. Due to the difficulty of visualising joint histograms globally, linear sensitivities are determined from the joint histograms ($P(\alpha|\tau_a)$) by weighting by the present day aerosol distribution (see Gryspeerdt et al., 2016). The first three subplots show the albedo sensitivity through modifying the $N_d$ (constant CF and LWP; Fig. 7a), CF (constant LWP; Fig. 7b) and AMSR-E LWP (constant CF; Fig. 7c). Both changes in $N_d$ and CF increase the scene albedo, which results in a negative radiative forcing. They have somewhat different spatial

15   patterns, with the albedo sensitivity to $N_d$ changes being concentrated in the centres of the stratocumulus decks due to the high liquid cloud fraction. The sensitivity to CF changes is highest at the edges of the stratocumulus decks, where the greatest



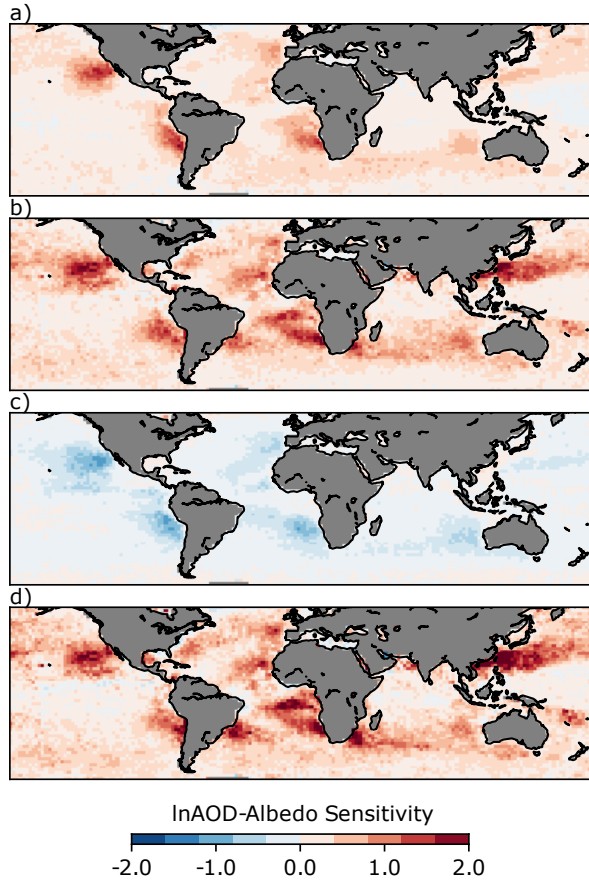

**Figure 7.** The sensitivity of cloud albedo to aerosol variations (a linear sensitivity calculated from $P(\alpha|\tau_a)$) through a) $N_d$ changes (Twomey only), b) CF changes (const. $N_d$ and LWP) and c) LWP changes. d) shows the total sensitivity, which is calculated directly using Eq. 2, not as a linear sum of a-c.

potential for modifying the cloud fraction exists, as found in previous studies (Gryspeerdt et al., 2016; Christensen et al., 2017; Andersen et al., 2017).

    The sensitivity to LWP changes is also strongly dependent on the liquid CF and so is strongest in the centres of the stratocumulus decks (Fig. 7c). As a reduction in LWP with increasing $N_d$ is observed in these regions (Fig. 3), this results in a negative albedo sensitivity to aerosol through LWP changes, which would in turn create a positive radiative forcing. The radiative forcing from these LWP changes (Fig. 7c) offsets 62% of the RFaci (Fig. 7a), resulting in a weakening of the ERFaci. This is likely the upper bound on the fraction of the RFaci offset by LWP reductions, following the results of Sec. 5 and supported by the weaker offsetting in regions with larger aerosol perturbations (e.g. the East China sea, the tropical and north Atlantic). Despite the reduced albedo sensitivity due to the LWP reduction, the overall albedo sensitivity to aerosols is still positive (Fig. 7d), re-





sulting in a negative ERFaci from liquid clouds due to the strong implied forcing from the $N_d$-CF relationship (approximately a 200% increase above the RFaci).

There remains considerable uncertainty in the magnitudes of these effects. The albedo change is only calculated over ocean. Observational studies suggest the $N_d$ change and RFaci over land are small, but it is possible that the LWP adjustments could

have a very different character and relationship to the RFaci over land. The variation in the $N_d$-LWP relationship in the Kilauea volcanic plume (Fig. 5) and the response of the LWP in shiptracks (Fig. 6), suggest that the LWP change determined in Fig. 7 is overly strong. This would then place a 62% offset of the RFaci as the upper bound on the radiative forcing from LWP changes (larger offsets are unlikely). This is consistent with previous work, where an increase in cloud albedo is found in response to a change in aerosol (Lebsock et al., 2008; Chen et al., 2014; Christensen et al., 2016), such that a LWP reduction cannot

completely offset the RFaci.

## 7 Discussion

This work demonstrates that a non-linear relationship exists between $N_d$ and LWP (Fig. 2). These results are in agreement with previous studies, with an increase in LWP with $N_d$ at low $N_d$ from precipitation suppression (E1a), but a decrease at high $N_d$ due to increased cloud top entrainment (E1b, c). The similarity in the relationship when using different measures of LWP

suggests that this relationship is not primarily due to LWP retrieval errors (E2). There are global variations in the $N_d$-LWP relationship and significant changes accompany variations in meteorological factors, particularly $RH_{750}$ (Fig.4). The observed $N_d$-LWP relationship implies a reduction in LWP with increasing aerosol and $N_d$, resulting in a positive radiative forcing that offsets around 60% of the RFaci.

The analysis in Sec. 5 suggests that the negative $N_d$-LWP relationship observed over much of the world may be overesti-

mated, resulting in too strong a corresponding positive radiative forcing due to aerosol induced LWP adjustments. A precipitation feedback (E3a) would produce a positive $N_d$-LWP relationship and so is unlikely to be responsible. An entrainment-based feedback on the $N_d$ (E3b) or an additional confounder (E4) could be responsible for the negative $N_d$-LWP relationship.

The albedo sensitivity to aerosol via LWP changes is particularly strong in the stratocumulus regions (Fig. 7), due to the high liquid cloud fraction. This implies an important role for the sedimentation-entrainment feedback (E1b). With the entrainment

of dry environmental air at the cloud top, the assumptions in the $N_d$ retrieval of a linearly increasing liquid water content and vertically constant $N_d$ no longer hold as the cloud is no longer adiabatic, such that the cloud top $N_d$ is no longer representative of the cloud base $N_d$. A reduction in the cloud top $r_e$ by homogeneous mixing during entrainment would produce an increase in $N_d$ required by E3b. Cloud top homogeneous mixing generating a apparent $N_d$-LWP would also create the dependence of the $N_d$-LWP relationship on $RH_{750}$ observed in Fig. 4. A stronger impact on the retrieved $N_d$ would be found with the entrainment

of drier air, resulting in a more negative $N_d$-LWP relationship.

However, although some studies have found evidence of homogeneous mixing in stratocumulus cloud (Breon and Doutriaux-Boucher, 2005; Yum et al., 2015), many studies have found that inhomogeneous mixing dominates, particularly at cloud top (Pawlowska et al., 2000; Gerber et al., 2005; Burnet and Brenguier, 2007; Yum et al., 2015). While inhomogeneous mixing





reduces the $N_d$, in extreme cases it does not result in a $r_e$ change, so may not be detected by satellite. As such, some proportion of homogeneous mixing is required for E3b to generate a negative $N_d$-LWP relationship in satellite data. A discrepancy between satellite retrieved and in-situ $N_d$ as a function of humidity or entrainment rate might be one indicator of this process. Further investigation into the mixing and behaviour of these retrievals at cloud top is necessary to establish the impact of E3b on the

$N_d$ retrievals and the $N_d$-LWP relationship.

An additional, unknown confounder (E4) is also a possible explanation for the results in Sec. 5. This effect would have to act on both $N_d$ and LWP together – A process that only affects one would not generate the systematic bias required. Even if such an unknown, additional confounding process exists, the conclusion drawn from Sec. 5 would still hold – that the implied aerosol impact on LWP in Fig. 7 is likely too strong.

Although volcanic emissions (Fig. 5) and shiptracks are exogeneous sources of aerosol, the datasets linked to these sources are limited. They occur in relatively restricted locations on the globe and there are a small number of the high $N_d$ retrievals required to populate the $N_d$-LWP histogram (Fig.6). While the shiptrack dataset is concentrated in stratocumulus region (Christensen et al., 2014), it is still possible that the effect on shallow cumulus clouds could be large enough to overcome the relatively small CF in this regime which has previously been shown to restrict the contribution of shallow cumulus clouds to the RFaci

(Gryspeerdt and Stier, 2012). Given the importance of the $N_d$ to this work, an improved understanding of the behaviour of the $N_d$ retrieval through a comparison with in-situ data is particularly important. Future studies are planned to expand this dataset of exogeneous aerosol perturbations in marine clouds such that a more representative global study of this type can be performed. Process-resolving simulations of these cases and a comparison to the global results are necessary to fully understand the behaviour of the satellite retrievals and how accurately they can represent the aerosol-$N_d$-LWP system to better constrain

the aerosol impact on LWP.

## 8  Conclusions

Along with liquid cloud fraction (CF) and droplet number concentration ($N_d$), the liquid water path (LWP) has a large impact on the albedo of a scene containing liquid clouds. However, due to the nature of the $N_d$-LWP relationship and the retrievals of these properties, global constraints of the aerosol impact on LWP and the corresponding radiative impact have been difficult to

determine. Several possible mechanisms for generating a relationship between $N_d$ and LWP are described in Sec. 2.

This work has demonstrated that although there is a clear relationship between the satellite-retrieved $N_d$ and LWP, this relationship is highly non-linear. At low $N_d$ values (where precipitation is expected), there is an increase in LWP with increasing $N_d$ consistent with an aerosol suppression of precipitation (E1a). At high $N_d$, the LWP decreases with increasing $N_d$, an effect which has been previously suggested to be due to the droplet size impact on entrainment (E1b/c, Fig.2). This non-linearity of

the $N_d$-LWP relationship restricts the ability of linear regressions to characterising the relationship. The reduction in LWP with increasing $N_d$ is only slightly stronger when using MODIS LWP compared to the in-cloud LWP from AMSR-E, suggesting that although correlated errors in the MODIS LWP and $N_d$ can play a role (E2), they do not dominate the magnitude of the $N_d$-LWP relationship.





By clustering the $N_d$-LWP joint histograms, it is shown that the primary variation in the histograms comes from variations in the LWP behaviour at high $N_d$ (Fig. 3). In the subtropical subsidence regions, there is a clear LWP reduction with increasing $N_d$, whilst in other regions, LWP remains constant or even increases with LWP even at high $N_d$. The global relationship is dominated by the subtropical relationship due to the high liquid CF in these regions, but the regional variations in the $N_d$-LWP relationship make it difficult to use the results from one region to constrain others.

Part of this variability come from regional differences in meteorological conditions. Significant variations in the $N_d$-LWP relationship are found with variations in $RH_{750}$ and LTS (Fig. 4). As with the global relationships, linear regressions have difficulty fully characterising these relationships. As noted by Chen et al. (2014) and Michibata et al. (2016), cloud top relative humidity plays an important role in determining the strength of the relationship, with a more weakly negative $N_d$-LWP relationship in humid regions.

However, results from natural experiments created by volcanic outgassing and shipping suggest that the negative $N_d$-LWP relationship is likely overestimated. In situations where the strong aerosol variability is the leading control on $N_d$ variations, the impact of feedbacks (E3) or additional confounders (E4) on the $N_d$-LWP relationship is significantly reduced. This suggests that the weaker $N_d$-LWP relationship observed in response to ship and volcanic aerosol perturbations better represents the impact of aerosols (E1) than the strong relationship observed at a global scale (Sec. 4), bringing the observations into better agreement with LES simulations Ackerman et al. (2004); Bretherton et al. (2007); Xue et al. (2008).

The observed $N_d$-LWP relationship suggests that LWP adjustments could offset up to 60% of the RFaci/Twomey effect (Fig. 7), as a positive radiative forcing. This represents an upper bound on the positive radiative forcing expected from a LWP reduction, as the results from natural experiments suggest that the LWP response is likely weaker than this (Figs. 5, 6). Further work is required to bound the LWP response, but these results suggest that the overall ERFaci is likely to be negative, supported by previous studies that have found a complete offset of the RFaci is unlikely (Chen et al., 2014).

Although it has been demonstrated in this work that the $N_d$-LWP relationship has a substantial impact on the ERFaci, it is clear that significant uncertainties remain. The satellite retrieved $N_d$-LWP relationship has several features that are similar to the relationship predicted by high resolution models (Ackerman et al., 2004; Sato et al., 2018), but it is not clear the extent to which these relationships represent the causal relationship and so can be used to constrain aerosol-cloud interactions. A wider study of the effect of aerosols on LWP due to exogenous aerosol perturbations in a variety of cloud regimes would provide one avenue for progress, as would finding a suitable mediating variable within the $N_d$-LWP relationship.

## Appendix A: Expected sensitivities

If the LWP and $N_d$ are calculated from MODIS data using the adiabatic assumption (Wood, 2006; Quaas et al., 2006), they take the form





$$N_d = 1.67 \times 10^{-8} c(T) f_{ad} \tau_c^{\frac{1}{2}} r_e^{-\frac{5}{2}} \tag{A1}$$

$$L = \frac{5}{9} f_{ad} r_e \tau_c \tag{A2}$$

$$\tag{A3}$$

where $0 < f_{ad} \leq 1$ is the adiabatic factor ($f_{ad}$=1 is completely adiabatic) and c(T) is the temperature correction to the condensation rate from Gryspeerdt et al. (2016). The linear sensitivity $\frac{d \ln L}{d \ln N_d}$ expected from $r_e$ variations, assuming a constant $\tau_c$ is then

$$\left.\frac{dL}{dN_d}\right|_{\tau_c} = \left.\frac{\partial L}{\partial r_e}\right|_{\tau_c} \left.\frac{\partial r_e}{\partial N_d}\right|_{\tau_c} \tag{A4}$$

$$\left.\frac{dN_d}{dr_e}\right|_{\tau_c} = -\frac{5}{2}\frac{N_d}{r_e} \tag{A5}$$

$$\left.\frac{dL}{dN_d}\right|_{\tau_c} = \frac{L}{r_e} \times -\frac{2}{5}\frac{r_e}{N_d} \tag{A6}$$

$$\left.\frac{d \ln L}{d \ln N_d}\right|_{\tau_c} = -\frac{2}{5} \tag{A7}$$

By similar logic, the sensitivity expected at a constant $r_e$ from variations in $\tau_c$ is

$$\left.\frac{d \ln L}{d \ln N_d}\right|_{r_e} = 2 \tag{A8}$$

Note that the cause of these variations is not specified. A variation in $r_e$ due to retrieval errors or $N_d$ variations would produce the same effect. As both the LWP and $N_d$ relate to the adiabatic factor in the same way as the optical depth, the expected sensitivity from adiabatic factor variation is also 2.

*Acknowledgements.* The MODIS data are from the NASA Goddard Space Flight Center. This work was supported by funding from the European Research Council under the European Union's Seventh Framework Programme (FP7/2007-2013) / ERC grant agreement no. FP7-306284 (QUAERERE). EG is supported by an Imperial College London Junior Research Fellowship. TG received funding from the European Union Horizon 2020 research and innovation program under the Marie Sklodowska-Curie grant agreement 703880.





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
