# Peer review of "Constraining the aerosol influence on cloud liquid water path"

_Atmospheric Chemistry and Physics, 2018_

## Referee Comment (RC1) · Anonymous Referee #1 · 2 Oct 2018

**Review of: Constraining the aerosol influence on cloud liquid water path**

By Gryspeerdt et. al.

This paper examines the response of LWP to changes in droplet number concentration (Nd) using long term (3 years), global observations. The results demonstrate a non-monotonic response with an increase in LWP under low Nd conditions followed by a negative trend under high Nd conditions. The trend flips approximately when precipitation is expected to be suppressed, so the authors conclude that the positive trend is due to rain suppression while the negative one is due to increase in entrainment. They also show that the results are sensitive to the RH of the environment and less sensitive to the LTS. Next, the authors use "natural experiments" of volcanic eruptions and ship track to better understand the causality of the trend. The radiative forcing due to the changes in LWP are calculated and shows that the reduction is LWP can, at most, cancel about half of the cooling due to changes in CF and cloud albedo.

In my opinion, the paper presents an innovative and impressive analysis of the data. A special effort was carried out to avoid artefacts and measurements errors. In addition, the topic is of high importance and hence I strongly recommend it for publication in ACP.

I do have suggestions and comments that the authors may want to consider:

**General comments**

- Another possible explanation for the positive correlation between Nd and LWP under low Nd conditions, that was proposed in the past (beside the rain suppuration that is mentioned in this paper) is warm cloud invigoration by aerosols. One way to separate the different causes using observations is to examining the effect of Nd on the cloud top height (CTH): The rain suppuration argument is expected to result in an increase in LWP without a significant change in CTH. However, the invigoration argument is expected to result in an increase in cloud vertical velocity, CTH and LWP at the same time. I think it could be interesting to examine it.

- For identifying the entrainment feedbacks, you use RH. However, what really determine evaporation and entrainment is the different in water vapor content between saturation (the cloud) and the environment. For a give RH this difference increases with increasing temperatures and hence may cause stronger evaporation and entrainment. For a small range of temperatures RH can serve as a good measure for the water vapor differences, but as here the analysis is conducted globally the range of temperatures are large and I think that this effect can interduce some errors and biases and can't be ignored. I think you should at least check its possible effect on the results and mention it in the text.

- I think that the argument that the "natural experiments" are the ground truth is not supported enough. It is true that it makes sense that the variations in Nd in the volcanic plume ,for example, were created by the aerosol concentration increase (and not by an artefact or any other reason) but you do not provide evidences that the meteorological conditions are really fixed. Can't the volcanic plum change other parameters that effect clouds (such as temperature and humidity) beside the aerosol concentration?

- All figures end at Nd~300 cm$^{-3}$. It doesn't sound very high. There aren't any cases with higher Nd?

- The abstract could be written in a clearer way (see specific comments below).

- The paper misses a few relevant previous papers.

**Specific comments**

- The first two sentences in the abstract are a bit confusing.

  "The **impact of aerosols on cloud properties** is one of the largest uncertainties in the anthropogenic radiative forcing of the climate. In recent years, significant progress has been made in constraining this forcing using observations, but uncertainty still remains, *particularly* in the **adjustments of cloud properties to aerosol perturbations**.".

  Both part discus the aerosol effect on cloud properties so the use of *"particularly"* here is not clear to me.

- P1, L11: "…suggesting that aerosol induced LWP reductions could offset a significant fraction of the radiative forcing from aerosol-cloud interactions (RFaci). "

  This sentence wasn't clear to me at the first time I read it. At this point you didn't define yet RFaci as an "instantaneous radiative forcing" so it is not clear why you do not consider the aerosol effect on LWP in "aerosol-cloud interactions"?

- The last sentence of the abstract is not clear to a reader that didn't read the paper yet.

- P1, L21: the change in CF or LWP can be caused not only by delay in precipitation but also by other reasons such as increase in evaporation and entrainment and warm cloud invigoration by aerosol.

- P2, L10: there were also previous studies that found a non-monotonic response of cloud properties (including LWP) to changes in aerosol concentration. The optimal aerosol concentration was shown to depends on the meteorological conditions.

- P2 L12: beside the meteorological conditions, the singe of the effect of aerosol on LWP may be determine by the range of changes in aerosol concentration that is examine in each case.

- P3 L2: see if you can write this part in a clearer way.

- P3: I thought that E1 b and c are more relevant in marine Sc and cumulus clouds, respectively. Is it correct? If yes, it is probably worth mentioning.

- P3: another pathway by which an increase in Nd may affect the LWP is by warm cloud invigoration. The increase in total droplet surface area under polluted conditions would lead to faster condensation (in the super saturated parts of the cloud,) more latent heat release, increase in cloud buoyancy and hence increase in LWP. In addition, under polluted conditions the smaller droplets would be pushed higher in the atmosphere (even under the same air vertical velocity). This could also lead to an increase in LWP with aerosol loading, as the clouds may reach higher in the atmosphere.

- Fig. 2: can you add maps presenting ml and mh for the different regions? I think it could be very interesting to see if the slopes (or even more interestingly the Nd that mark the change in trend) change in the different regions/meteorological conditions and whether you can identify that regions that support the development of more develop clouds are more effected by the increase in Nd than other regions.

- Figures: it will be interesting to add to the figures the Nd that differ between the two different slops. It is interesting to see if it changes for the different cases (presented in the different figures).

- P8. L12: it is consistent with **at list** two aerosol effects in liquid clouds. Under extremely clean conditions the clouds could be "aerosol limited" and so cloud invigoration was suggested to take place.

- P8. L32: another (simpler) way to overcome this difficulty would be to plot the Nd marking the change in trend or the slopes of the different trends on a map.

- Fig. 3: Do you think that it is possible that you don't see any significant trends for cluster 2 because it mixes many different regions with different meteorological conditions (i.e. tropics and extra-tropics)?

- P12 last line: is it possible that the volcano adds water vapor to the atmosphere as well as aerosols and hence you see a cancelation of effects? In other words, are you sure that all other meteorological conditions are on average the same between the years?

- P13 L13: I am not sure that the statement that the volcanic case has a: "reduced impact of other processes (E2-4)" is supported well enough. For example, are the meteorological condition really fixed? I can imagine that a volcanic plum has other

effects rather than increasing the aerosol loading (such as changes in the temperature or humidity vertical profile).

- P13 L29: are they distinguishable from 0?

- P14 L2-4: again, they are not reduced completely. I think you make this argument too strong.

- P16 L14: it doesn't have to be because of cloud top entrainment. It could also be due to increase in lateral entrainment.

---

## Referee Comment (RC2) · Anonymous Referee #2 · 21 Oct 2018

This manuscript is another valiant attempt to improve our understanding of whether the consequences of aerosol-cloud interactions ("aci") can be detected using satellite observations. There are major things I like in this paper: a fabulous dedicated section elucidating what processes may be actually happening during aci and what we may be seeing (or think we're seeing) from satellites instead; the hypothesis of the existence of two regimes where the aerosol effects on droplet numbers (Nd) and cloud evolution may be completely different depending on the base state of the cloud; and the potential use of cases of "natural experiments" to distinguish actual from perceived aerosol effects. However, I have a fundamental, philosophical uneasiness with using notoriously unreliably-retrieved cloud variables such as Nd and LWP as basis for the analysis. The authors try to ameliorate things by moving away from MODIS-based LWP (at least for

some of the analysis), but AMSR-E has its own issues and a mismatch in the reff and LWP retrieval scales is introduced. [Digression]: It always amazes me that from the reflectance of two MODIS channels one can retrieve four pieces of information, optical depth, reff, LWP and Nd. Yes, I understand they are related and this provides a weak (because of the many assumptions) constraint. But let's pull back and think about it: many combinations of droplet concentrations and sizes can give the same LWP and optical depth. There is a reason Nd is not included as a MODIS product, it's just too uncertain. When the first unofficial Nd products started to creep up, retrievals were performed on only overcast or near-overcast areas; now we started retrieving everywhere all the time, even if f_ad may be varying wildly. It is also unknown whether assumptions about linearly increasing LWC are better than the vertically constant LWC adopted in the official MODIS product. The authors are aware of most of these issues, as suggested by lines 16-23 of p. 3 (even though I should point out that the greatest worry is not random but systematic errors in OD and reff retrievals). Another problem is that Nd from MODIS corresponds to near cloud top (something that should have been disclosed earlier than the discussion section), while LWP is a vertically integrated quantity. In the context of aci does it make sense to correlate the two since aerosols will mostly affect Nd near the cloud base? I guess one implicitly assumes that Nd is constant with height, which then has implications about droplet size vertical profiles when LWC is increasing with height. I admit that I'm unsure whether all these caveats can alter the qualitative characteristics of the Nd-LWP histograms (or is it just a matter of shifting values in the same direction?) which are the centerpiece of the analysis, but I'm nevertheless uneasy with taking Nd and LWP retrievals at face value.

– Even though I have the same philosophical reservations I expressed here, this contemporaneous ACP paper may be worth taking a look at and perhaps citing, https://www.atmos-chem-phys-discuss.net/acp-2018-697/.

– As with many studies of this type, the authors should make clear that they do not examine the temporal co-evolution of Nd and LWP to understand how they interact/relate

10.5194/acp-2018-885
2018
Atmos. Chem. Phys. Discuss.

as individual clouds thicken, thin out, produce/suppress precipitation. They rather compare different (static) incidences or cloud snapshots at 1 degree of scales.

– Lines 3-4 of p.5: Are you using the QA flags of the retrievals at all? I believe MODIS identifies edge pixels. Given your selection/filtering method, what is the range of CF at $1°$ degree scales?

– p. 6, first paragraph: Are you served well by a single global histogram given systematic changes of SZA with latitude?

– p.6, lines 27-30: May be it's just me, but I don't understand what you're saying here. Perhaps it can be written more clearly.

– p. 9, lines 1-7: Cluster 1 seems to be more frequent in the tropics than Cluster 2, so I'm not sure that characterizing Cluster 1 as "subtropical" and Cluster 2 as occurring "mostly in the tropics and extratropics" is accurate. Confusion is furthered by essentially calling Cluster 2 a low liquid-CF cluster, and also the main cluster of the Malavelle et al. (2017) study which I don't believe looked at low liquid-CF clouds. Are you sure that two clusters are sufficient to describe the diversity of Nd-LWP histograms?

– Fig. 7a shows, I believe, what has been previously called "cloud susceptibility" (Platnick and Twomey 1994; Oreopoulos and Platnick 2008) and it's a missed opportunity to not identify it as such.

– p. 18. Lines 3-5: Mid-latitude storm tracks also have very high CFs. Your mainly Cluster 2 southern oceans are covered by overcast supercooled liquid clouds.

---

## Author Comment (AC1) · 18 Jan 2019

**Response to reviewer 1**
* * *
*: This paper examines the response of LWP to changes in droplet number concentration (Nd) using long term (3 years), global observations. The results demonstrate a non-monotonic response with an increase in LWP under low Nd conditions followed by a negative trend under high Nd conditions. The trend flips approximately when precipitation is expected to be suppressed, so the authors conclude that the positive trend is due to rain suppression while the negative one is due to increase in entrainment. They also show that the results are sensitive to the RH of the environment and less sensitive to the LTS. Next, the authors use "natural experiments" of volcanic eruptions and ship track to better understand the causality of the trend. The radiative forcing due to the changes in LWP are calculated and shows that the reduction is LWP can, at most, cancel about half of the cooling due to changes in CF and cloud albedo. In my opinion, the paper presents an innovative and impressive analysis of the data. A special effort was carried out to avoid artefacts and measurements errors. In addition, the topic is of high importance and hence I strongly recommend it for publication in ACP. I do have suggestions and comments that the authors may want to consider:*

**Reply**: We thank the reviewer for their comments and address each of them in turn below.

**General Comments**
* * *
*: Another possible explanation for the positive correlation between Nd and LWP under low Nd conditions, that was proposed in the past (beside the rain suppuration that is mentioned in this paper) is warm cloud invigoration by aerosols. One way to separate the different causes using observations is to examining the effect of Nd on the cloud top height (CTH): The rain suppuration argument is expected to result in an increase in LWP without a significant change in CTH. However, the invigoration argument is expected to result in an increase in cloud vertical velocity, CTH and LWP at the same time. I think it could be interesting to examine it.*

**Reply**: We thank the reviewer for pointing out the additional hypothesis of warm cloud invigoration, which we now discuss in the revised manuscript. We would suggest that it is not clear that precipitation suppression would lead to no change in cloud top height. Pincus and Baker (1994) calculate an increase in the equilibrium cloud thickness as a function of $N_d$, suggesting that an increase in CTH is not enough to conclusively differentiate between precipitation suppression and warm cloud invigoration as the process responsible for an increase in LWP. Fig. R1 shows how the LWP depends on the cloud geometrical properties determined by CALIOP. There is a strong relationship between LWP and cloud geometrical thickness, as would be expected by a (sub-) adiabatic cloud model. However, there is not a strong link between $N_d$ and the cloud geometrical thickness. However, we note that cloud geometrical thickness might be one

[Figure]

Figure R1: $N_d$ and LWP from MODIS as a function of the cloud geometrical properties, as determined by CALIOP for the Peruvian stratocumulus deck (100W-80W, 10S-30S) for the years 2007-2011. This diagonal lines are contours of cloud top altitude. Cloud base is determined by CALIOP where possible, CloudSat otherwise - this has little effect on the overall pattern of the results. The data is from the CCCM product (Kato et al., 2010).

way to disentangle the $N_d$-LWP relationship and plan to look at this in more in depth in the future.
* * *
*: For identifying the entrainment feedbacks, you use RH. However, what really determine evaporation and entrainment is the different in water vapor content between saturation (the cloud) and the environment. For a give RH this difference increases with increasing temperatures and hence may cause stronger evaporation and entrainment. For a small range of temperatures RH can serve as a good measure for the water vapor differences, but as here the analysis is conducted globally the range of temperatures are large and I think that this effect can interduce some errors and biases and can't be ignored. I think you should at least check its possible effect on the results and mention it in the text.*
**Reply**: Thank you for pointing this out. We have repeated Fig. 4,e separating by high and low (q-q$_{sat}$) at 750hPa (Fig. R2). While the separation between high and low saturation deficit is somewhat stronger, we prefer to keep the relative humidity figure for continuity with previous work (especially Ackerman et al., 2004). The additional figure will be referenced and placed in supplementary information.
* * *
*: I think that the argument that the "natural experiments" are the ground truth is not supported enough. It is true that it makes sense that the variations in Nd in the volcanic plume ,for example, were created by the aerosol concentration*

[Figure]

Figure R2: As Fig. 4, but using saturation deficit instead of RH

*increase (and not by an artefact or any other reason) but you do not provide evidences that the meteorological conditions are really fixed. Can't the volcanic plum change other parameters that effect clouds (such as temperature and humidity) beside the aerosol concentration?*

**Reply**: Previous studies found the island effect a negligible component of signal in the cloud properties downwind of volcanoes (Ebmeier et al., 2014). Other studies have shown that the aerosol is the important component in shiptracks, rather than the water produced by combustion (Hobbs et al., 2000). While Kilauea has a significant thermal power (around $1\,\mathrm{GW}$; Wright and Flynn, 2004), for a narrow plume only $100\,\mathrm{km}$ across and $500\,\mathrm{m}$ thick, it heats the plume by less than $0.01\,^{\circ}\mathrm{C}$, assuming a $5\,\mathrm{ms}^{-1}$ windspeed. Similar arguments suggest that any water production from the volcano would not have a significant effect on the clouds downwind. While this small effect is the case for an effusive eruption (such as that in 2008) an explosive eruption would be considerably more energetic and could have a much larger effect on the local meteorology.
* * *
*: All figures end at Nd 300 cm$^{-3}$. It doesn't sound very high. There aren't any cases with higher Nd?*

**Reply**: The upper limit is actually around $500\,\mathrm{cm}^{-3}$. With an optical depth of 4, this corresponds to cloud top effective radius of around $5\mu\mathrm{m}$. Although smaller effective radii are possible, this is at the limits of the retrieval and we feel that the large values of $N_d$ are likely to be highly error prone. As such, they are excluded from this work.
* * *
*: The abstract could be written in a clearer way (see specific comments below).*
**Reply**: The abstract has been modified following the comments below.
* * *
*: The paper misses a few relevant previous papers.*
**Reply**: The reference list has been expanded and now in particular also includes references to the "warm invigoration" hypothesis.

**Specific Comments**
* * *
*: The first two sentences in the abstract are a bit confusing. "The impact of aerosols on cloud properties is one of the largest uncertainties in the anthropogenic radiative forcing of the climate. In recent years, significant progress has been made in constraining this forcing using observations, but uncertainty still remains, particularly in the adjustments of cloud properties to aerosol perturbations." Both part discus the aerosol effect on cloud properties so the use of "particularly" here is not clear to me.*

**Reply**: "Adjustments" here refers to the changes in cloud properties that are not an instantaneous forcing, following the terminology from the last IPCC report (Boucher et al., 2014). These sentences have been modified for improved clarity.
* * *
**P1, L11:**: *"suggesting that aerosol induced LWP reductions could offset a significant fraction of the radiative forcing from aerosol-cloud interactions (RFaci).*

" This sentence wasn't clear to me at the first time I read it. At this point you didn't define yet RFaci as an "instantaneous radiative forcing" so it is not clear why you do not consider the aerosol effect on LWP in "aerosol-cloud interactions"?

**Reply**: "Instantaneous" is now specified here to make it clearer. The term RFaci is used following Boucher et al. (2014).
* * *
*: The last sentence of the abstract is not clear to a reader that didn't read the paper yet.*

**Reply**: This sentence has been modified for clarity
* * *
**P1, L21:**: *the change in CF or LWP can be caused not only by delay in precipitation but also by other reasons such as increase in evaporation and entrainment and warm cloud invigoration by aerosol.*

**Reply**: Thank you for pointing this out, we have modified this sentence and included a discussion of warm cloud invigoration as E1d.
* * *
**P2, L10:**: *there were also previous studies that found a non-monotonic response of cloud properties (including LWP) to changes in aerosol concentration. The optimal aerosol concentration was shown to depends on the meteorological conditions.*

**Reply**: We have added these further references here and later in the paper where the non-monotonic response is discussed.
* * *
**P2 L12:**: *beside the meteorological conditions, the singe of the effect of aerosol on LWP may be determine by the range of changes in aerosol concentration that is examine in each case.*

**Reply**: The paragraph beginning E1 now mentions that these effects may depend on the local meteorological and aerosol environment
* * *
**P3 L2:**: *see if you can write this part in a clearer way.*
**Reply**: Amended
* * *
**P3:**: *I thought that E1 b and c are more relevant in marine Sc and cumulus clouds, respectively. Is it correct? If yes, it is probably worth mentioning.*

**Reply**: We would suggest that it is not clear which processes dominate at this stage and so have presented them as hypotheses. The results in the volcanic plume suggest that E1b and c may not dominate to the extent previously thought.
* * *
**P3:**: *another pathway by which an increase in Nd may affect the LWP is by warm cloud invigoration. The increase in total droplet surface area under polluted conditions would lead to faster condensation (in the super saturated parts of the cloud,) more latent heat release, increase in cloud buoyancy and hence increase in LWP. In addition, under polluted conditions the smaller droplets would be pushed higher in the atmosphere (even under the same air vertical velocity). This could also lead to an increase in LWP with aerosol loading, as the clouds*

*may reach higher in the atmosphere.*
**Reply**: Added in as E1d
* * *
**Fig. 2:**: *can you add maps presenting ml and mh for the different regions? I think it could be very interesting to see if the slopes (or even more interestingly the Nd that mark the change in trend) change in the different regions/meteorological conditions and whether you can identify that regions that support the development of more develop clouds are more effected by the increase in Nd than other regions.*
**Reply**: Applying the fit to individual gridboxes is tricky as many locations do not adequately fill the whole $N_d$-LWP space in a way that allows the fit to be applied without a large error. For example, in some regions, only a negative relationship is observed, as low $N_d$ retrievals do not occur often enough to fill in that part of the histogram. As such, the low $N_d$ part of the piecewise function is almost unconstrained. The clustering method is explicitly designed to deal with such a situation, as it fills in missing regions of the histogram with the values from the nearest cluster. While this makes some assumptions about the behaviour of the $N_d$-LWP histogram in locations where it is not fully specified, it suffices for showing the variation in the relationship globally. The first sentence of the regional results section is modified to highlight this.
* * *
**Figures:**: *it will be interesting to add to the figures the Nd that differ between the two different slops. It is interesting to see if it changes for the different cases (presented in the different figures).*
**Reply**: We are not clear what the referee is requesting here
* * *
**P8. L12:**: *it is consistent with at list two aerosol effects in liquid clouds. Under extremely clean conditions the clouds could be "aerosol limited" and so cloud invigoration was suggested to take place.*
**Reply**: Amended
* * *
**P8. L32:**: *another (simpler) way to overcome this difficulty would be to plot the Nd marking the change in trend or the slopes of the different trends on a map.*
**Reply**: This is addressed above.
* * *
*: Fig. 3: Do you think that it is possible that you don't see any significant trends for cluster 2 because it mixes many different regions with different meteorological conditions (i.e. tropics and extra-tropics)?*
**Reply**: This is possible, but we consider it unlikely. As shown in the meteorological separation section, even under restricted variations in meteorological conditions, positive relationship are rarely observed at high $N_d$, suggesting that they are not being offset by negative relationships.
* * *
**P12 last line:**: *is it possible that the volcano adds water vapor to the atmosphere as well as aerosols and hence you see a cancelation of effects? In other words, are you sure that all other meteorological conditions are on average the same between the years?*
**Reply**: This is addressed above

**P13 L13:**: *I am not sure that the statement that the volcanic case has a: "reduced impact of other processes (E2-4)" is supported well enough. For example, are the meteorological condition really fixed? I can imagine that a volcanic plum has other effects rather than increasing the aerosol loading (such as changes in the temperature or humidity vertical profile).*
**Reply**: See above
* * *
**P13 L29:**: *are they distinguishable from 0?*
**Reply**: There are some values of the in-track $N_d$ (around $100\text{cm}^{-3}$) where the dLWP is distinguishable from 0. More data would be required to be certain.
* * *
**P14 L2-4:**: *again, they are not reduced completely. I think you make this argument too strong.*
**Reply**: We agree that these other effects (E3, E4) are not completely removed. However, we feel that the inferences made still stand. The impact of the volcano on the thermal and humidity structure of the atmosphere is minimal, suggesting that E4 is largely accounted for (although it is clearly difficult to be sure as it includes possible "unknown unknowns"). The exogeneous aerosol variations from the ships and volcano ensure that E3a is not an issue as it has not had time to act. While E3b could still affect our results, its impact is strongly reduced. As it would be expected to produce a reduction in LWP with increasing LWP, this would mean that our result is likely a lower bound on the reduction in LWP expected from an increase in $N_d$/aerosol. An extra clause has been added to point out that E3 and E4 are not completely removed.
* * *
**P16 L14:**: *it doesn't have to be because of cloud top entrainment. It could also be due to increase in lateral entrainment.*
**Reply**: Amended

**Response to reviewer 2**
* * *
*: This manuscript is another valiant attempt to improve our understanding of whether the consequences of aerosol-cloud interactions (aci) can be detected using satellite observations. There are major things I like in this paper: a fabulous dedicated section elucidating what processes may be actually happening during aci and what we may be seeing (or think we're seeing) from satellites instead; the hypothesis of the existence of two regimes where the aerosol effects on droplet numbers (Nd) and cloud evolution may be completely different depending on the base state of the cloud; and the potential use of cases of "natural experiments" to distinguish actual from perceived aerosol effects. However, I have a fundamental, philosophical uneasiness with using notoriously unreliably-retrieved cloud variables such as Nd and LWP as basis for the analysis. The authors try to ameliorate things by moving away from MODIS-based LWP (at least for some of the analysis), but AMSR-E has its own issues and a mismatch in the reff and LWP retrieval scales is introduced. [Digression]: It always amazes me that from*

*the reflectance of two MODIS channels one can retrieve four pieces of informa-
tion, optical depth, reff, LWP and Nd. Yes, I understand they are related and
this provides a weak (because of the many assumptions) constraint. But let's
pull back and think about it: many combinations of droplet concentrations and
sizes can give the same LWP and optical depth. There is a reason Nd is not
included as a MODIS product, it's just too uncertain. When the first unofficial
Nd products started to creep up, retrievals were performed on only overcast
or near-overcast areas; now we started retrieving everywhere all the time, even
if $f_{ad}$ may be varying wildly. It is also unknown whether assumptions about
linearly increasing LWC are better than the vertically constant LWC adopted
in the official MODIS product. The authors are aware of most of these issues,
as suggested by lines 16-23 of p. 3 (even though I should point out that the
greatest worry is not random but systematic errors in OD and reff retrievals).*

**Reply**: We thank the reviewer for their comments and have addressed them in
turn below. A section on systematic biases has been added as E2c. We agree
that these are also important and could play a role in these results. We have
focussed on the random errors, as they would be present even if a cloud property
could be retrieved ideally. The sub-adiabatic factor represented the impact of
systematic biases in the previous version, but we agree it is more complete to
include a section on them separately.
* * *
*: Another problem is that Nd from MODIS corresponds to near cloud top
(something that should have been disclosed earlier than the discussion section),
while LWP is a vertically integrated quantity. In the context of aci does it
make sense to correlate the two since aerosols will mostly affect Nd near the
cloud base? I guess one implicitly assumes that Nd is constant with height,
which then has implications about droplet size vertical profiles when LWC is
increasing with height. I admit that I'm unsure whether all these caveats can
alter the qualitative characteristics of the Nd-LWP histograms (or is it just a
matter of shifting values in the same direction?) which are the centerpiece of
the analysis, but I'm nevertheless uneasy with taking Nd and LWP retrievals at
face value.*

**Reply**: We agree that this is a tricky problem, unfortunately it is not clear
that there is currently a better way to deal with these issues. This is the main
reason that we feel that the results presented here are only able to bound the
aerosol impact on LWP. We are hopeful that future improved retrievals of these
properties will lead to a stronger constraint on these cloud processes.
* * *
*: Even though I have the same philosophical reservations I expressed here, this
contemporaneous ACP paper may be worth taking a look at and perhaps citing,
https://www.atmos-chem-phys-discuss.net/acp-2018-697/.*

**Reply**: Thank you for drawing our attention to this paper, we have included it
in the discussion of the results. We note that our preliminary work suggest that
the sub-grid $N_d$-LWP relationship may be different to the relationship at larger
spatial and temporal scales. The cause of this difference is not yet certain, but
if the interpretation of the results from the natural experiments is correct, these

relationships determined at small spatial and temporal scales may be primarily due to effects other than E1, due to the lack of aerosol variation to drive the $N_d$ variation at these very small scales.
* * *
*: As with many studies of this type, the authors should make clear that they do not examine the temporal co-evolution of Nd and LWP to understand how they interact/relate as individual clouds thicken, thin out, produce/suppress precipitation. They rather compare different (static) incidences or cloud snapshots at 1 degree of scales.*
**Reply**: A sentence highlighting this has been added to the methods section
* * *
**Lines 3-4 of p.5***: Are you using the QA flags of the retrievals at all? I believe MODIS identifies edge pixels. Given your selection/filtering method, what is the range of CF at 1 degree scales?*
**Reply**: The standard MODIS cloud optical properties product already excludes cloud edge pixels. We remove further pixels close to the cloud edge by selecting only cases where the 5km cloud fraction is greater than 0.9, meaning that in the worse case, the average closest cloud is almost 2km for each 5km pixel. This restricts the fraction of pixels used for the analysis, but the $1°$ cloud/utilised pixels fraction still varies between 0 and 100%, even if having 100% utilised pixels is rare compared to the standard MODIS product.
* * *
**p. 6, first paragraph:***: Are you served well by a single global histogram given systematic changes of SZA with latitude?*
**Reply**: This is a good point, as it is known that there are SZA biases in the retrieval. A sentence in the results section has been included to highlight this. However, the use of the global histograms is used primarily to compare to the use of a single linear regression. For this use, we believe that a single global histogram is sufficient.
* * *
**p.6, lines 27-30:***: May be it's just me, but I don't understand what you're saying here. Perhaps it can be written more clearly.*
**Reply**: The argument is that the forcing number itself is highly sensitive to the anthropogenic fraction used, so it is not particularly useful. By comparing the forcing from LWP changes to the RFaci calculated using the same anthropogenic fraction, a more useful metric is obtained. A similar enhancement of the RFaci is obtained when using a different anthropogenic fraction product. This result is now included in the results section.
* * *
**p. 9, lines 1-7:***: Cluster 1 seems to be more frequent in the tropics than Cluster 2, so I'm not sure that characterizing Cluster 1 as "subtropical" and Cluster 2 as occurring "mostly in the tropics and extratropics" is accurate. Confusion is furthered by essentially calling Cluster 2 a low liquid-CF cluster, and also the main cluster of the Malavelle et al. (2017) study which I don't believe looked at low liquid-CF clouds. Are you sure that two clusters are sufficient to describe the diversity of Nd-LWP histograms?*
**Reply**: This section has been modified to address these points. The figure has

been changed to better show where there is no data. Cluster 1 is described as being the in "subtropical subsidence regions", which is consistent with their position around 20N/S. Cluster 2 is now referred to as "dominates in the tropics and mid-latitudes". The references to "low liquid CF" have been changed "larger ice CF". As noted, two clusters may not completely describe the variability in the histograms, but they are sufficient for showing that there is global variation in the histogram. This is now noted in the first paragraph of this section.
* * *
*: Fig. 7a shows, I believe, what has been previously called "cloud susceptibility" (Platnick and Twomey 1994; Oreopoulos and Platnick 2008) and it's a missed opportunity to not identify it as such.*

**Reply**: It is a similar property to the cloud susceptibility. The cloud susceptibility is the relationship between cloud albedo and $N_d$ at constant LWP. In Fig. 7a, the cloud susceptibility is multiplied by the $N_d$ sensitivity to AOD, which results in a slightly different property of the cloud field, as it also has a dependence on the aerosol activation.
* * *
**p. 18. Lines 3-5:***: Mid-latitude storm tracks also have very high CFs. Your mainly Cluster 2 southern oceans are covered by overcast supercooled liquid clouds.*

**Reply**: A fair point, this has been modified to point out the covariance of liquid CF with high $N_d$ in the cluster 1 regions.

**Bibliography**

Ackerman, A. S., Kirkpatrick, M. P., Stevens, D. E., and Toon, O. B.: The impact of humidity above stratiform clouds on indirect aerosol climate forcing, Nature, 432, 1014, https://doi.org/10.1038/nature03174, 2004.

Boucher, O., Randall, D. A., Artaxo, P., Bretherton, C., Feingold, G., Forster, P. M., Kerminen, V.-M., Kondo, Y., Liao, H., Lohmann, U., Rasch, P., Satheesh, S. K., Sherwood, S., Stevens, B., and Zhang, X. Y.: Clouds and Aerosols, https://doi.org/10.1017/CBO9781107415324.016, 2014.

Ebmeier, S. K., Sayer, A. M., Grainger, R. G., Mather, T. A., and Carboni, E.: Systematic satellite observations of the impact of aerosols from passive volcanic degassing on local cloud properties, Atmos. Chem. Phys., 14, 10 601–10 618, https://doi.org/10.5194/acp-14-10601-2014, 2014.

Hobbs, P. V., Garrett, T. J., Ferek, R. J., Strader, S. R., Hegg, D. A., Frick, G. M., Hoppel, W. A., Gasparovic, R. F., Russell, L. M., Johnson, D. W., O'Dowd, C., Durkee, P. A., Nielsen, K. E., and Innis, G.: Emissions from Ships with respect to Their Effects on Clouds, J. Atmos. Sci., 57, 2570–2590, https://doi.org/10.1175/1520-0469(2000)057¡2570:EFSWRT¿2.0.CO;2, 2000.

Kato, S., Sun-Mack, S., Miller, W. F., Rose, F. G., Chen, Y., Minnis, P., and Wielicki, B. A.: Relationships among cloud occurrence frequency, overlap, and effective thickness derived from CALIPSO and CloudSat merged cloud vertical profiles, J. Geophys. Res., 115, D00H28, https://doi.org/10.1029/2009JD012277, 2010.

Pincus, R. and Baker, M.: Effects of precipitation on the albedo susceptibility of clouds in the marine boundary layer, Nature, p. 250, 1994.

Wright, R. and Flynn, L. P.: Space-based estimate of the volcanic heat flux into the atmosphere during 2001 and 2002, Geology, 32, 189, https://doi.org/10.1130/G20239.1, 2004.

---

## Author Response (AR2)

**Response to comments**

**Main comment**: *There is a tendency in this paper to take coarsely aggregated data and then explain them via detailed cloud process. There is a broader question of the causal relationship vs. the general association between LWP and $N_d$. The associations are implicitly assumed to be causal when you resort to physical explanations as opposed, e.g., to co-variability of $N_a$ and meteorological drivers that change LWP. This may fit into E4. At the very least, please point this out.*

**Reply**: Thank you for pointing this out. We have amended the end of E4 to include the possibility of aerosol-LWP covariations producing a $N_d$-LWP relationship. As for isolating the causal $N_d$-LWP relationship, we would suggest that the main argument of the paper is that the $N_d$-LWP retrieved by satellite is not the causal relationship, such that its use would imply too large and aerosol effect. The results in section 5 demonstrate that the $N_d$-LWP relationship is strongly influenced by other factors, especially where the aerosol variability is low.We would suggest that the $N_d$-LWP relationship downwind of the volcano and in the shiptrack regions is more likely to be causal. However, for this reason, the study itself can only provide a bound to the $N_d$ impact on LWP. The abstract and conclusions have been modified to better highlight this.
* * *
**Abstract**: *check for grammar:aerosol impact \*on\* LWP, Lines 9 and 10 : "in the relationship" repeated*
**Reply**: Amended
* * *
**Pg 3, line 25**: *Aren't tau_c and r_e biases correlated? E.g., in broken cloud fields, r_e is biased high and tau_c biased low?*
**Reply**: We agree that systematic biases are also likely an issue. Random biases are included first as the generation of relationships through random errors is perhaps more surprising. A mention of broken clouds has been included in E2c to highlight the importance of systematic biases.
* * *
**Pg 4, line 14**: *$N_d$ (not $N_D$)*
**Reply**: Amended
* * *
**Pg 6**: *Doesn't the fact that you give equal weight to all $N_d$ bins bias interpretation? If an $N_d$ doesn't occur frequently then why give it similar weight? (Consider the co-variability comment above.)*
**Reply**: We would argue that this is a benefit, rather than a flaw, as it better highlights the form of the relationship. It also has the benefit that the shape of the relationship doesn't change if the $N_d$ distribution changes (as would be expected due to anthropogenic aerosol emissions). This is now noted in the manuscript.
* * *
**Pg 8, line 5**: *You have done a nice job differentiating between the evaporation-entrainment feedback and the sedimentation-entrainment feedback, so why not*

*refer to Xue et al. (2008) or Xue and Feingold 2006 here too.*
**Reply**: These references have been included.
* * *
**Pg 9 line 32**: *see main comment. Here is one example. The Malavelle (2017) study shows that on average LWP is   constant. However there are large spatial differences across the domain. So I dont see how you can invoke Malavelle here.*
**Reply**: We have modified this sentence to weaken it slightly. It was meant to state that a weak sensitivity of LWP to $N_d$ is not unexpected, rather than that the $N_d$ impact on LWP is small.
* * *
**Pg 12, line 13**: *same comments as Pg 8, line 5.*
**Reply**: These references are now included.
* * *
**Pg 13, line 7 and line 10**: *Have you actually shown that the meteorological conditions are similar across the years or do you surmise?*
**Reply**: We had not shown that the meteorological conditions are the same, basing the argument on the similarity of the upwind (region B) histograms in the two years. While there is a difference in LTS of around 1K both up and downstream of Hawaii, it seems unlikely that this could be responsible for the change downwind of the island if there is no change upwind. This is now noted.
* * *
**Pg 15, line 28 and Pg 16, line 5**: *I may not have been paying sufficient attention but I didn't understand the 62%*
**Reply**: This offset is based on the calculated values of the forcing from the RFaci and the LWP adjustment. The absolute values are not stated, only the relative offset, as this highlights the cloud processes involved in the LWP adjustment, removing uncertain components (such as the anthropogenic aerosol fraction) that impact both the RFaci and the LWP adjustment. The wording has been modified here and the explanation improved in. An extra example using a different anthropogenic fraction has also been included to demonstrate the independence of the offset.
* * *
**Fig. 7**: *Is this dln albedo/dln AI or d albedo/dln AI?*
**Reply**: The albedo is linear, but AOD is logarithmic. The figure and caption have been modified to make this clearer.
* * *
**Pg 17, line 18**: *"an" apparent*
**Reply**: Amended
* * *
**Pg 17, line 23**: *an early, and beautiful observational paper is Nicholls 1987? (QJRMS)*
**Reply**: many thanks, this has now been included.
* * *
**Pg 19**: *I may be wrong on this but I seem to recall doing this calculation and showing $N_d$ dependent on sqrt($f_{ad}$). Boers and Mitchell (1994) might have shown similar. In any case, I don't remember it being linear. Worth a quick check.*
**Reply**: Thank you for pointing this out. A1 has now been amended so that $N-d$ is dependent on sqrt($f_{ad}$) and sqrt(c(Y)).

[revised manuscript text omitted]